# Food Addiction and Lifestyle Habits among University Students

**DOI:** 10.3390/nu13041352

**Published:** 2021-04-18

**Authors:** Cristina Romero-Blanco, Antonio Hernández-Martínez, María Laura Parra-Fernández, María Dolores Onieva-Zafra, María del Carmen Prado-Laguna, Julián Rodríguez-Almagro

**Affiliations:** Department of Nursing, Physiotherapy and Occupational Therapy, Ciudad Real Faculty of Nursing, University of Castilla-La Mancha, 13071 Ciudad Real, Spain; cristina.romero@uclm.es (C.R.-B.); marialaura.parra@uclm.es (M.L.P.-F.); mariadolores.onieva@uclm.es (M.D.O.-Z.); carmina.prado@uclm.es (M.d.C.P.-L.); julianj.rodriguez@uclm.es (J.R.-A.)

**Keywords:** food addiction, Yale Food Addiction Scale, body mass index, sleeping behaviors, physical activity, sedentary behavior, Mediterranean diet, anxiety, depression, lifestyle

## Abstract

The prevalence of overweight and obesity is increasing in our society, with a complex, multifactorial origin, and associated with greater morbidity and mortality in the population. Food addiction (FA) is a common disorder in overweight/obese people, which appears to be increasingly common in young people. This study analyzed food addiction in a group of young university students and to examine its association with body composition, quality of sleep, adherence to the Mediterranean diet, physical activity/sedentary habits, tobacco or alcohol consumption, and health status. A total of 536 undergraduate nursing students participated in a questionnaire that included the Yale Food Addiction Scale (YFAS 2.0). Up to 6.4% of the students presented FA. Statistically significant associations were observed in the variables for sleep quality odds ratio (OR) 4.8 (95% confidence interval (CI): 1.66–13.87), anxiety/depression OR 8.71 (95% CI: 3.93–19.27), body mass index (BMI) OR 8.32 (95% CI: 3.81–18.15) and sedentary lifestyle OR 2.33 (95% CI: 1.09–5.01). A predictive model was developed after binary logistic regression (area under the ROC curve 0.84 (95% CI: 0.77–0.91). Students with FA presented higher BMI values, worse sleep quality, anxiety or depression problems, and more time spent in sedentary behaviors.

## 1. Introduction

Over the last 50 years, the prevalence of overweightness and obesity worldwide has increased considerably, with dietary changes and sedentary lifestyles being considered some of the main underlying causes of this pandemic [1]. The World Health Organization (WHO) defines overweightness and obesity as abnormal or excessive fat accumulation that may impair health. The WHO global has estimated that globally 39% of adults (people aged 18 years and over) were overweight and around 13% of the world’s adult population are obese. The prevalence of obesity is increasing in most European countries, with a projected prevalence of obesity of between 13% and 43% by 2025 [2]; with 33 of the 53 countries projected to have an obesity prevalence of 20% or more. Ireland and Italy are predicted to have the highest (43%) and the lowest (13%) prevalence, respectively.

Overweight and obesity have been associated with multiple diseases such as type 2 diabetes mellitus, dyslipidemia, reproductive disorders, certain types of cancer and hypertension, in addition to a considerable economic burden associated with addressing these obesity-related pathologies [3]. Both genetic and non-genetic factors are related to higher body weight. Some of the non-genetic factors, such as eating behavior, can modulate the genetic predisposition to obesity through epigenetic mechanisms. In fact, genetic associations with obesity have been found to be stronger among those who have a higher consumption of certain high-fat foods or sugary beverages [4,5]. Therefore, interventions to combat obesity need to address irregular eating habits as one of the main causes of overweight and obesity [6].

Today’s society consumes many foods for the sake of pleasure, and not only because of the need for sustenance. Even in the absence of hunger, the environment around us causes us to consume certain foods because they appear in advertising campaigns or because of previous satisfactory experiences we have had [7]. This urge to eat certain foods has been called food addiction and is described as a form of eating that focuses on foods that are highly palatable and pleasurable. This pleasure associated with eating these types of foods affects the reward circuits in the same way as other addictive substances such as alcohol or opioids. This has generated some debate as to what the exact name of this disorder should be, since only certain foods cause this addiction and not any type of food [8].

The fifth edition of the Diagnostic and Statistical Manual of Mental Disorders (DSM-5) establishes a section on addictive disorders, which, as a novelty, includes gambling addiction; however, food addiction, as well as other types of addictive disorders, are still not included [9]. The debate regarding the inclusion of food addiction is still open and scientific articles studying this disorder have increased in recent years [10]. The Yale Food Addiction Scale [11] is the measurement system most commonly used by researchers to evaluate food addiction which establishes an assessment considering the criteria of the Diagnostic and Statistical Manual of Mental Disorders (DSM). An update was performed in 2016 (YFAS 2.0) [12] due to changes in the diagnostic criteria for addictive disorders included in this manual.

Many of the studies that have evaluated food addiction have focused primarily on assessing patients who have overweight/obesity or other mental disorders [13,14] finding that there is a relationship between food addiction and overweight, anxiety, depression, binge eating disorders, anorexia and bulimia. However, it has been observed that in the non-clinical population, young people are highly influenced by the food environment and food addiction is more prevalent in the age group between 18 and 29 years old [15].

It would be interesting to study this population to determine the prevalence of overweight/obesity, analyze their healthy and unhealthy habits, and to evaluate whether there is a relationship with food addiction. Therefore, the aim of this study was to analyze food addiction in a group of university students and to assess their health habits, body mass index and health status. Specifically, we aimed to evaluate the incidence of food addiction in the university setting and to determine whether it is related to students’ body mass index, physical activity level, type of diet, quality of sleep, smoking habits, alcohol consumption and health status.

## 2. Materials and Methods

### 2.1. Participants

A total of 620 nursing students from the University of Castilla-La Mancha were invited to participate in this study, of whom 536 agreed. This study was approved by the ethics committee of the Hospital General Universitario de Ciudad Real (code c291-register 11/2019).

Data were collected through a questionnaire that included sociodemographic questions, and items related to health status and lifestyle habits in terms of diet, physical activity, quality of sleep, smoking and alcohol consumption.

### 2.2. Measures

#### 2.2.1. Demographics

Data were collected regarding sex, age, height, and weight by means of an ad-hoc questionnaire as well as questions regarding tobacco use (yes/no), and alcohol consumption (yes/no). Body mass index (BMI) was obtained from the formula BMI = weight(kg)/(height(m))^2^. BMI was classified into three types: normal weight (18.5–24.9 kg/m^2^), underweight and overweight/obese.

#### 2.2.2. Dietary Behaviors

The Mediterranean diet adherence questionnaire (PREDIMED) [16] was used to analyze diet. This questionnaire consists of 14 questions, each of which scores between 0 and 1 point. The analysis yields a score of good (9 or more points) or poor adherence to the Mediterranean diet.

#### 2.2.3. Physical Activity and Sedentary Behaviors

The International Physical Activity Questionnaire (IPAQ) short version was used for the analysis of physical activity [17]. This enabled us to determine the weekly minutes of moderate and vigorous physical activity (MVPA) performed by the participants. The World Health Organization recommends at least 150 min of MVPA per week [18]. The physical activity variable was evaluated as “yes” or “no” based on compliance or non-compliance with the physical activity recommendations. Hours spent in sedentary behavior were also evaluated. This variable was dichotomized based on a threshold of 6 h or less of daily sedentary behavior. This threshold was calculated taking the daily hours of class (hours of obligatory sedentary activity) and the mean of the sample into account.

#### 2.2.4. Quality of Sleep

The Pittsburgh Sleep Quality Index (PSQI) was used (Cronbach’s α = 0.72) [19]. This questionnaire contains 19 items with a final score between 0 and 21 points. The self-rated items of the PSQI generate seven component scores (with subscales ranged 0–3): sleep quality, sleep latency, sleep duration, habitual sleep efficiency, sleep disturbance, use of sleeping medication, and daytime dysfunction. Scores less than or equal to 5 were classified as “good sleep quality”.

#### 2.2.5. Health Status

Health status was assessed using the European Quality of Life scale (EQ5D) [20]. We used the question related to anxiety/depression from this questionnaire. Participants who presented some or many problems were considered as a “yes” for the variable “anxiety/depression” versus those who did not present any problems.

#### 2.2.6. Food Addiction

The Food Addiction Scale (YFAS 2.0) (Cronbach’s α = 0.94) [21] consists of 35 questions scored on an 8-level Likert scale ranging from 0 (never) to 7 (every day). Each question is scored as present/absent based on a threshold determined in the YFAS 2.0 validation document. The questionnaire provides a score based on 12 diagnostic criteria and allows for two scoring methods: symptom count or diagnostic threshold. For the “diagnosis” scoring option, a participant can be evaluated as having mild (2 or 3 symptoms and clinical significance), moderate (4 or 5 symptoms and clinical significance) or severe food addiction (6 or more symptoms and clinical significance). For the symptom count scoring option only the first 11 diagnostic criteria are taken into account and therefore the scores range from 0 to 11. Criterion number 12 (clinically significant impairment or distress) is taken into account for the “diagnosis” score.

### 2.3. Statistical Analysis

First, descriptive statistics were performed using absolute and relative frequencies for categorical variables, while the mean and standard deviation (SD) were used for quantitative variables.

Subsequently, a bivariate analysis was performed to determine the relationship between the different sociodemographic characteristics, dietary habits, physical activity, sleep quality and health status with food addiction using Pearson’s Chi-square test and estimating the odds ratio (OR) with its respective 95% confidence interval (CI).

Finally, a multivariate analysis was carried out using binary logistic regression between all factors and food addiction. The objective of this analysis was to develop a predictive model of food addiction, using the backward stepwise variable selection system of SPSS. With the variables selected in the multivariate model, the predictive capacity was calculated using the area under the ROC curve (AUC). In order to assess the prediction in qualitative terms, the Swets’ criteria, whose values range from 0.5–0.6 (bad), 0.6–0.7 (poor), 0.7–0.8 (satisfactory), 0.8–0.9 (good), and 0.9–1.0 (excellent), was used [22].

## 3. Results

A total of 79.2% (408) of participants were women and the prevalence of overweight or obesity was 10.3% (53). Table 1 shows the descriptive characteristics of the sample.

The mean number of food addiction symptoms in the sample was 0.94 with a standard deviation of 1.77. Regarding the diagnosis of food addiction in terms of diagnostic threshold, 33 students (6.4%) presented food addiction (mild 1.7% (9), moderate 1.9% (10) and severe 2.7% (14)).

Additionally, the 12 symptoms were also assessed to determine which was more present in each diagnostic group. Table 2 shows the number and percentage of students who presented each of the symptoms, separated into both food addiction and addiction levels. Students who did not present food addiction scored mainly on symptom 1 (15.6%) (substance taken in larger amount and for longer period than intended); those with mild addiction presented mainly symptom 4 (44.4%) (important social, occupational, or recreational activities given up or reduced) and 8 (44.4%) (continued use despite social or interpersonal problems), for moderate addiction, participants scored for symptom 7 (70.0%) (characteristic withdrawal symptoms; substance taken to relieve withdrawal) and in the case of severe addiction, symptom 2 was most present (92.2%) (persistent desire or repeated unsuccessful attempts to quit).

Next, a bivariate analysis was performed between sociodemographic characteristics and factors that characterize lifestyle and food addiction (Table 3). In the diagnosis of food addiction, there were no significant associations in gender, alcohol, tobacco, diet and physical activity. Statistically significant associations were observed in the variables of sleep quality (*p* = 0.001), anxiety/depression (*p* < 0.001), BMI (*p* < 0.001) and sedentary lifestyle (*p* = 0.020). Overall, students with food addiction presented worse sleep quality, a certain amount of anxiety or depression, overweight or obesity and more sedentary behavior.

Thereafter a predictive model of risk of food addiction was developed. In this multivariate analysis, the predictor variables for food addiction were overweight/obesity with an OR of 8.08 (95% CI: 3.35–19.51), sedentary lifestyle greater than 6 h with an OR of 2.44 (95% CI: 1.04–5.71), anxiety or depression with an OR of 7.79 (95% CI: 3.29–18.42) and sleep quality, with an OR of 3.24 (95% CI: 1.04–10.13). The ROC AUC (Figure 1) of this model was 0.84 (95% CI: 0.77–0.91), which is considered a good predictive capacity, according to Swets’ criteria.

## 4. Discussion

This study has analyzed the sociodemographic situation and health habits of 536 university nursing students in relation to food addiction. The findings reveal a 6.4% prevalence of food addiction. In addition, associations were found with sleep quality, BMI, anxiety/depression, and sedentary lifestyle. However, no associations were observed with adherence to the Mediterranean diet, physical activity, sex, and tobacco smoking or alcohol consumption. The predictive model for food addiction risk was considered good with sleep, BMI, anxiety/depression, and sedentary behavior being predictor variables for this addiction. To our knowledge, this is the first study to analyze food addiction with this tool in a group of undergraduate nursing students.

The prevalence of food addiction among the students analyzed was 6.4% (0.94 points). These data are lower than those found in other studies that analyzed large cohorts of nonclinical populations of all ages [23]. In general, there is great controversy regarding the prevalence of food addiction depending on the population studied [24], although in the non-clinical population it ranges from 8.2% to 22.2%. Nonetheless, there is agreement regarding its high prevalence in people with high BMI, eating disorders and binge eating disorders. Age also appears to be an important factor since high scores have been found in children and adolescents, as well as in young adults [15,25]. In our case, we found a low prevalence of food addiction among students. This could be related to the training received since studies similar to our sample, such as the one by Fawzi et al. in a sample of medical students found scores of 0.93 [26]. Likewise, Aloi et al. [27] conducted a study among Italian medical students, reporting values somewhat lower than ours.

### 4.1. Overweight/Obesity

Food addiction may be involved in the pathogenesis of obesity [28], in fact, some authors suggest that weight loss may also exert an effect on food addiction by decreasing food addiction [29]. The prevalence of food addiction in obese patients ranges from 16.5% to 40% and may even be higher if the patient also has some other type of eating disorder [28]. In this study, the prevalence of food addiction among overweight/obese students was 42.4% and, in line with previous studies, relationships between food addiction and BMI were also found. Many of the research studies conducted so far have included patients with high body mass indexes [24,30,31] demonstrating the close relationship with food addiction. Jimenez-Murcia et al. [29] suggest that people with food addiction could be classified into three clusters with different severity and treatment implications; the mildest cluster would include mostly obese people, compared to clusters 1 and 2 where there would be more people with other eating disorders and where interventions should be more complex and include pharmacological treatment. These data, together with those found in our study, suggest that in this population with high BMI levels and with food addiction, and even without addiction, the therapeutic approach should focus on lifestyle modification: providing tools to improve sleep quality, strategies to reduce anxiety and improve quality of life and reduce sedentary behavior.

### 4.2. Anxiety/Depression

Food addiction and its positive association with other mental disorders has been widely demonstrated, particularly its relationship with eating disorders such as anorexia, bulimia, or binge eating disorder, but also with anxiety and depression [13]. To our knowledge, the European Quality of Life instrument (Euroqol) has not been used in other studies of food addiction. This questionnaire accounts for both anxiety and depression, revealing whether the participant presents any problems related to these disorders. The results obtained in this study support the positive relationship with food addiction, in line with Borisenkov et al. [32] who carried out the analysis of food addiction and its relationship with psycho-emotional elements in a large sample of Russian university students. Both results confirm the relationship of these parameters in non-clinical populations.

### 4.3. Sleep Quality

Another element frequently related to food addiction is sleep quality. University students with food addiction also have poorer sleep quality [33]. Night-time binge eating has also been shown to be related to food addiction, although it seems that this relationship is stronger in the general population than in university students; however, in both groups it is associated with poor sleep quality [34]. These data suggest that sleep disturbances may lead to increased impulsive behaviors and that food addiction disorders could be addressed through improved sleep quality.

### 4.4. Physical Activity and Sedentary Lifestyles

In our study, food addiction was associated with extended sedentary behavior, but not with the physical activity performed. When performing the analysis, we accounted for whether or not participants met the WHO recommendations for physical activity, however, we also analyzed the total minutes of physical activity with food addiction and failed to find an association. The WHO recommendations only consider the amount of moderate vigorous activity. Previous studies suggest that people with food addiction spend less time performing MVPA and more time engaged in sedentary behavior [35]. This relationship has not been analyzed in depth in young university students. It would be necessary to objectively examine (by means of accelerometers) the amount of physical activity performed. Furthermore, it would be interesting to explore whether there is any relationship with addictive behaviors towards sport, as it has been observed that in athletes the conjunction of both addictions exists [36]. Young university students may be trying to compensate for their addiction to food by over-exercising, which in turn may lead to a dependence on exercise.

### 4.5. Smoking and Alcohol

No relationship between smoking or alcohol and food addiction was found in this study. The relationship between tobacco smoking and alcohol with food addiction is controversial. Some researchers have found a higher prevalence of food addiction among the population that is addicted to drugs, tobacco, or alcohol [37], although there was no evidence of associations between each of the substances and food addiction. Other authors did find certain associations for alcohol, cannabis, and tobacco in adolescents, although with small effect sizes [38]. Pipová et al. also found a relationship between food addiction and tobacco addiction as well as tobacco use in a nonclinical population [39] finding differences only between nonsmokers and social smokers, although not for regular smokers. In a population similar to that of our study, Najem et al. [33] found a relationship with food addiction in university smokers and ex-smokers.

Among the main strengths observed in this article, we highlight the fact that this is the first study to have developed a predictive model for food addiction with good predictive capacity. Moreover, validated tools have been used with extensive experience in their use.

This study presents a series of limitations. The first is due to the fact that the participants recruited are university nursing students and we are unaware of whether these results may vary among students from other disciplines. Secondly, there may be errors in data collection due to the fact that self-administered questionnaires were used. However, these were validated questionnaires. Another problem is that since few cases of food addiction were detected, it was not possible to perform the analysis between the three levels of addiction. For the same reason it was also not possible to separate overweight/obese students in the analysis. It would have been interesting to assess whether the associations detected were more prevalent in one group or another. Finally, it would also be appropriate to validate this model in a different population to determine whether the predictive capacity is maintained.

## 5. Conclusions

The aim of the study was to identify which lifestyle factors of nursing students were related to food addiction. Our results found no association with smoking or alcohol consumption and also no association with physical activity or Mediterranean diet. However, relationships were found with sedentary lifestyle, anxiety/depression and sleep. These lifestyle elements could be the focus of behavior modification for students to decrease the prevalence of food addiction.

The results of the present study could suggest that food addiction is not the origin of the remaining problems, but rather the consequence of the confluence of various habits. Additionally, it allows us to think about whether treatments based on behavioral modifications could be effective and additional pharmacological or other treatments might not be necessary. Prevention of unhealthy lifestyles could therefore prevent the onset of food addiction and its treatment. Previous studies have shown that by controlling excess weight or improving the quality of sleep, the number of people diagnosed with food addiction could be reduced [29,33], which supports our hypothesis. In this study, findings are the result of a single measurement over time, and it is very daring to establish the origin of a problem based on these results. The design of the study does not allow us to find evidence that food addiction is the origin or the consequence of these habits. Obviously, this is only a hypothesis that could be put forward.

University nursing students, due to their training, receive knowledge regarding the effects on health or disease as a consequence of habits. This knowledge could improve control over their habits. The predictive model has shown a higher risk of food addiction in variables that are related to some habits. It could be the reason why a lower prevalence of food addiction has been found compared to other non-clinical population groups. The acquisition of health knowledge is not the solution to the problem of food addiction, since there are other factors acquired in previous years that affect the final result. However, health education could be the first step towards initiating a change of habits.

It would be interesting to begin interventions with this population, throughout their undergraduate education, to include motivational aspects and knowledge of healthy habits in terms of reducing sedentary lifestyles, improving sleep quality or minimizing stress or anxiety. This would enable us to determine whether there is a change in BMI and food addiction rates.

## Figures and Tables

**Figure 1 nutrients-13-01352-f001:**
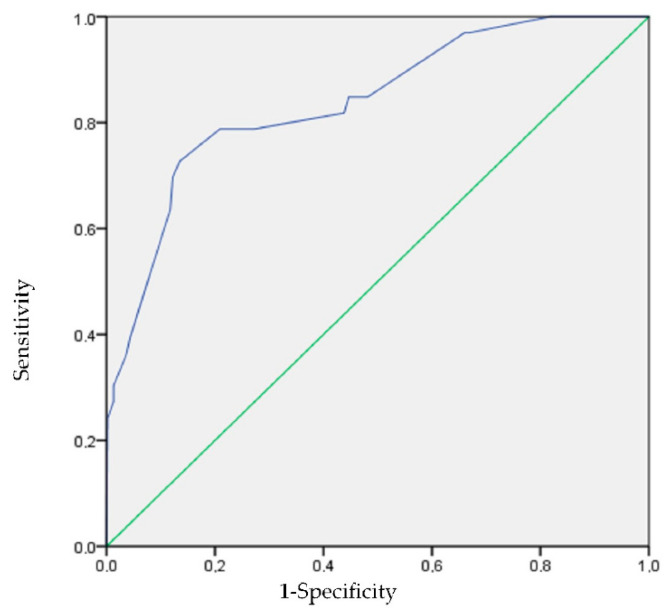
ROC curve. Area under the ROC curve to determine the predictive ability of the model, representing the sensitivity on the ordinate axis and 1-specificity in the abscissa.

**Table 1 nutrients-13-01352-t001:** Descriptive variables.

	*N* (%)	Mean (SD)
Gender		
Woman	408 (79.2)	
Man	107 (20.8)	
Mediterranean diet		7.51 ± 1.90 (a)
Good adherence	154 (29.9)	
Poor adherence	361 (70.1)	
Physical activity		236.80 ± 349.82 (b)
More than 150 min/week	231 (44.9)	
Less than 150 min/week	284 (55.1)	
Sedentarism		6.53 ± 3.04 (c)
Less than or equal to 6 h	242 (49.0)	
More than 6 h	252 (51.0)	
BMI		22.36 ± 7.02
Normal weight	411 (80.1)	
Underweight	49 (9.6)	
Overweight/obesity	53 (10.3)	
Sleep		5.75 ± 2.98 (d)
No sleeping difficulties	196 (38.1)	
Has sleeping difficulties	319 (61.9)	
Anxiety/depression		
No problem	378 (73.4)	
Some/many problems	137 (26.6)	
Smoking habits		
Smoker	61 (11.8)	
Non-smoker	454 (88.2)	
Alcohol		
Doesn’t drink alcohol	95 (18.4)	
Drinks alcohol	420 (81.6)	

(a) PREDIMED score: from 0 to 14 points. (b) Minutes per week; (c) hours per day; (d) Pittsburgh score: from 0 to 21.

**Table 2 nutrients-13-01352-t002:** Symptoms of food addiction: descriptive table.

	No Food Addiction *N* (%)	Food Addiction *N* (%)		
		Mild	Moderate	Severe
YFAS Symptom 1				
Substance taken in larger amount and for longer period than intended	75 (15.6)	2 (22.2)	5 (50)	12 (85.7)
YFAS Symptom 2				
Persistent desire or repeated unsuccessful attempts to quit	49 (10.2)	3 (33.3)	6 (60.0)	13 (92.2)
YFAS Symptom 3				
Much time/activity to obtain, use, recover	32 (6.6)	2 (22.2)	6 (60.0)	10 (71.4)
YFAS Symptom 4				
Important social, occupational, or recreational activities given up or reduced	23 (4.8)	4 (44.4)	5 (50.0)	10 (71.4)
YFAS Symptom 5				
Use continues despite knowledge of adverse consequences	24 (5)	2 (22.2)	4 (40.0)	10 (71.4)
YFAS Symptom 6				
Tolerance	13 (2.7)	3 (33.3)	4 (40.0)	11 (78.6)
YFAS Symptom 7				
Characteristic withdrawal symptoms; substance taken to relieve withdrawal	24 (5)	0 (0)	7 (70.0)	10 (71.4)
YFAS Symptom 8				
Continued use despite social or interpersonal problems	18 (3.7)	4 (44.4)	3 (30.0)	10 (71.4)
YFAS Symptom 9				
Failure to fulfill major role obligation	16 (3.3)	1 (11.1)	1 (10.0)	7 (50.0)
YFAS Symptom 10				
Use in physically hazardous situations	32 (6.6)	0 (0.0)	1 (10.0)	8 (57.1)
YFAS Symptom 11				
Craving, or a strong desire or urge to use	5 (1.0)	1 (11.1)	2 (20.0)	7 (50.0)
YFAS Symptom 12				
Use causes clinically significant impairment or distress	6 (1.2)	9 (100.0)	10 (100.0)	14 (100.0)

YFAS: Yale Food Addiction Scale.

**Table 3 nutrients-13-01352-t003:** Bivariate and multivariate analysis.

	Food Addiction		*p* Value (X^2^)	OR 95% CI	* aOR 95% CI
	No *N* (%)	Yes *N* (%)			
Gender					
Woman	381 (93.4)	27 (6.6)	0.452	1 (ref.)	
Man	101 (94.4)	6 (5.6)		0.84 (0.34–2.09)	
BMI					
Normal weight	394 (95.9)	17 (4.1)	**<0.001**	1 (ref.)	1 (ref.)
Underweight	47 (95.9)	2 (4.1)		0.99 (0.22–4.40)	0.96 (0.20–4.56)
Overweight/obese	39 (73.6)	14 (26.4)		**8.32 (3.81–18.15)**	**8.08 (3.35–19.51)**
Mediterranean diet					
Good adherence	144 (93.5)	10 (6.5)	0.548	1 (ref.)	
Poor adherence	338 (93.6)	23 (6.4)		0.98 (0.46–2.11)	
Physical activity					
More than 150 min/week	215 (93.1)	16 (6.9)	0.399	1 (ref.)	
Less than 150 min/week	267 (94.0)	17 (6.0)		0.86 (0.42–1.73)	
Sedentarism					
Less than or equal to 6 h	232 (95.9)	10 (4.1)	**0.020**	1 (ref.)	1 (ref.)
More than 6 h	229 (90.9)	23 (9.1)		**2.33 (1.09–5.01)**	**2.44 (1.04–5.71)**
Sleep					
No sleeping difficulties	192 (98.0)	4 (2.0)	**0.001**	1 (ref.)	1 (ref.)
Has sleeping difficulties	290 (90.9)	29 (9.1)		**4.8 (1.66–13.87)**	**3.24 (1.04–10.13)**
Anxiety/depression					
No problem	369 (97.6)	9 (2.4)	**<0.001**	1 (ref.)	1 (ref.)
Some/many problems	113 (82.5)	24 (17.5)		**8.71 (3.93–19.27)**	**7.79 (3.29–18.42)**
Smoking habits					
Non-smoker	426 (93.8)	28 (6.2)	0.350	1 (ref.)	
Smoker	56 (91.8)	5 (8.2)		1.36 (0.50–3.66)	
Alcohol					
Doesn’t drink alcohol	87 (91.6)	8 (8.4)	0.248	1 (ref.)	
Drinks alcohol	395 (94.0)	25 (6.0)		0,69 (0.30–1.58)	

OR: odds ratio; CI: confidence interval; Bold: *p* < 0.05; * multivariate analysis; ref: reference.

## Data Availability

The data presented in this study are available on request from the corresponding author.

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
