# Peer review of "Food Addiction and Lifestyle Habits among University Students"

_nutrients, 2021, doi:10.3390/nu13041352_

Round 1

Reviewer 1 Report

This is an interesting study that aims to evaluate the incidence of food addiction in the university setting and to determine body composition, physical activity/sedentary behavior, type of diet, quality of sleep, smoking habits, alcohol consumption, and health status. The paper is well-written and the data are exhaustively presented.

Author Response

Response to reviewer #1

This is an interesting study that aims to evaluate the incidence of food addiction in the university setting and to determine body composition, physical activity/sedentary behavior, type of diet, quality of sleep, smoking habits, alcohol consumption, and health status. The paper is well-written and the data are exhaustively presented.

Thank you very much for your comment. We are very pleased that you appreciated our manuscript.

Reviewer 2 Report

 Romero-Blanco present data on food addiction and its association with lifestyle and anxiety depression in polish nursing students.  They find that food addiction is associated with sedentary habits, anxiety/depression, higher BMI and lower sleep quality. They go on to develop a predictive model for risk of food addiction, The data is collected and analyzed appropriately. However, the low number of individuals with FA preclude a more detailed analysis of the stratification in this subpopulation and the underlying differences between levels of FA. There are occasional problems with language and the paper will benefit from a second round of editing. A few grammatic and other minor errors are highlighted below:

In table 1, N and percentage for alcohol does not add up.

 Add more descriptive legend for figure 1.

 From Line 180 on the text reads like discussion. Is this subtitle missing?

Line 12: Overweight people instead of “people with overweight”?

Line 13: analyzed instead of “aims to analyze”.

Line 51: Suggest shortening to “be, since only certain foods cause this addiction”.

Line 74: Physical activity levels instead of “physical activity/sedentary behavior”

Author Response

Response to reviewer #2

Romero-Blanco present data on food addiction and its association with lifestyle and anxiety depression in polish nursing students.  They find that food addiction is associated with sedentary habits, anxiety/depression, higher BMI and lower sleep quality. They go on to develop a predictive model for risk of food addiction, The data is collected and analyzed appropriately. However, the low number of individuals with FA preclude a more detailed analysis of the stratification in this subpopulation and the underlying differences between levels of FA. There are occasional problems with language and the paper will benefit from a second round of editing. A few grammatic and other minor errors are highlighted below:

 Thank you very much for your comments. Your review helps us to improve our manuscript.

In table 1, N and percentage for alcohol does not add up.

Thank you for your comment. Table 1 has been corrected.

Add more descriptive legend for figure 1.

Figure legend: Area under the ROC curve to determine the predictive ability of the model, representing the sensitivity on the ordinate axis and 1-specificity in the abscissa.

 From Line 180 on the text reads like discussion. Is this subtitle missing?

Thank you for your comment. It has been corrected.

Line 12: Overweight people instead of “people with overweight”?

Thank you for your comment. It has been corrected.

Line 13: analyzed instead of “aims to analyze”.

Thank you for your comment. It has been corrected.

Line 51: Suggest shortening to “be, since only certain foods cause this addiction”.

Thank you for your comment. It has been corrected.

Line 74: Physical activity levels instead of “physical activity/sedentary behavior”

Thank you for your comment. It has been corrected.

Reviewer 3 Report

This was an interesting paper that explored food additction prevalence and a predictive model of factors associated with food addiction in a higher education population, specifically nurses. Overall there are a few points that need further clarification.

  1. The introduction of the paper is focused on obesity and the prevalence of obesity but in the title and in other sections of the paper (including the abstract) you have focused in overweight. This is misleading to the readers. It would be helpful if you included in the introduction some information about the link to obesity (definition) and prevalence of overweight.
  2. In line 43 you state that today's society consumes foods for the sake of a pleasure, evidence for this?
  3. Lines 52-56: Although the authors state that food addiction should be included in the DSM 5 and that views of these are mixed, there is no clarity why this section adds to the narrative of the study and whether this study can add to this narrative. 
  4. Lines 89-91: Why are overweight/obese people grouped together? Do the authors have any evidence that you can treat these two groups the same? 
  5. Lines 109-110: Can you provide a clarification that the values suggested for good sleep quality are according to the scale. 
  6. When looking at the symptom prevalence across population was this assessed statistically or just descriptive? If this was done statistically, please report the actual test/values. If not, I would recommend to do this statistically, as it can be a bit misleading or to make this quite clear.
  7.  Lines 153-161: The authors state that "In the diagnosis of food addiction, there were no significant differences between men and women" Was this does as a specific test? If yes report the test and values. Currently where it is placed it is misleading - The logistic regression that was conducted as stated in the analysis section was done to identify predictors of food addiction, as such when gender was entered, this should have been phrased as that gender was not a significant predictor of food addiction and not that there were no differences between the two genders. It is possible with a chi-square a significant difference might exist. Until that is done this will need to be rephrased. 
  8. Similarly for BMI, anxiety/depression and sleep quality, the authors phrase it about significant associations, but no correlations values or chi squares are reported. This is misleading as what they have found is that those variables are significant predictors of food addiction. If this is a misunderstanding on my part, then it would be good to clarify these section and for the authors to actually report the actual tests. Although when looking at the p values they match the ones on Table 3. 
  9. Add what is meant by 'ref' in Table 3 in the note section. 
  10. Line 180: separate section under a "discussion" heading
  11. section 4.2: please remove the line about not being able to compare the findings of current study with other studies that have measured anxiety/depression as no other study has used that measure. If this line is kept please provide evidence that the measure that you have used for depression and anxiety is significantly operationalising those two concepts differently than other measures of depression and anxiety. Please provide a justification in introduction and method why this particular measure was used instead of others. 
  12. Please include cronbachs alpha values where appropriate. 
  13. The conclusion does not match your findings. Where in the study did you acquire evidence that food addiction is not the origin of the remaining problems but rather the consequence of the confluence of various habits? This is not what you investigated. Generally you are over estimating what you results are telling you and please take into consideration the design of the study which is one measurement in time.  
  14. Overall the conclusion really needs to rewritten and reconsidered. Some of the sentences make no sense or have no actual points. 

Author Response

Response to reviewer #3

This was an interesting paper that explored food additction prevalence and a predictive model of factors associated with food addiction in a higher education population, specifically nurses. Overall there are a few points that need further clarification.

  1. The introduction of the paper is focused on obesity and the prevalence of obesity but in the title and in other sections of the paper (including the abstract) you have focused in overweight. This is misleading to the readers. It would be helpful if you included in the introduction some information about the link to obesity (definition) and prevalence of overweight.

Thank you for your comment. The introduction has been modified: overweight and obesity have been defined and prevalence has been estimated. The introduction has also been modified for better understanding.

  1. In line 43 you state that today's society consumes foods for the sake of a pleasure, evidence for this?

Thank you for your comment.

Reference number 7 defines hedonic eating and explains further characteristics of hedonic feeding. More information can also be found in the following reference:

Lowe MR, Butryn MI. Hedonic hunger: a new dimension of appetite? Physiol Behav 2007;91:432–9.

  1. Lines 52-56: Although the authors state that food addiction should be included in the DSM 5 and that views of these are mixed, there is no clarity why this section adds to the narrative of the study and whether this study can add to this narrative. 

Thank you for your comment. This explanation was included as we are using the term addiction and it should be clear that it is not listed in the DSM 5 as a mental disorder. On the other hand, this has been included in the narrative as the instrument for measuring food addiction (YFAS 2.0) follows the criteria of this manual. Our study aims with this paragraph to justify the use of this questionnaire. If you feel that we should remove this paragraph, we will remove it.

  1. Lines 89-91: Why are overweight/obese people grouped together? Do the authors have any evidence that you can treat these two groups the same? 

They were grouped together due to the low prevalence in our population of students with BMI values above 25. Since the WHO defines both as BMI values that are detrimental to health, we decided to group them together. This has been included in the limitations section.

  1. Lines 109-110: Can you provide a clarification that the values suggested for good sleep quality are according to the scale. 

The first validation of the Pittsburgh scale in Spanish was in 1997 (Crombach alpha 0.81). This article explains that values less than or equal to 5 were classified as "good sleep quality". Manzar et al. (2016) also used this value in their study with university students. More information can also be found in the following references:

  • Royuela, A., & Macías, J. (1997). [Clinimetricas properties of the Spanish version of the questionnaire of Pittsburgh]. Vigilia-Sueño, 9, 81–94
  • Manzar, M.D.; Zannat, W.; Hussain, M.E.; Pandi-Perumal, S.R.; Bahammam, A.S.; Barakat, D.; Ojike, N.I.; Olaish, A.; Spence, D.W. Dimensionality of the Pittsburgh Sleep Quality Index in the collegiate young adults. Springerplus 2016, 5, doi:10.1186/s40064-016-3234-x.

  1. When looking at the symptom prevalence across population was this assessed statistically or just descriptive? If this was done statistically, please report the actual test/values. If not, I would recommend to do this statistically, as it can be a bit misleading or to make this quite clear.

Thank you for your comment. The symptom prevalence was just descriptive.

This was not correctly explained in the manuscript. It has been clarified both in the text and in the table.

Statistical analysis was not considered for inclusion as most of the population did not present any of the symptoms.

  1.  Lines 153-161: The authors state that "In the diagnosis of food addiction, there were no significant differences between men and women" Was this does as a specific test? If yes report the test and values. Currently where it is placed it is misleading - The logistic regression that was conducted as stated in the analysis section was done to identify predictors of food addiction, as such when gender was entered, this should have been phrased as that gender was not a significant predictor of food addiction and not that there were no differences between the two genders. It is possible with a chi-square a significant difference might exist. Until that is done this will need to be rephrased. 

Thank you for your comment. Table 3 shows the results obtained in the bivariate analysis (chi-square test) for all the variables studied. No significant differences were obtained for gender and therefore it was not included in the predictive model.

The sentence in the manuscript has been modified for a better understanding.

  1. Similarly for BMI, anxiety/depression and sleep quality, the authors phrase it about significant associations, but no correlations values or chi squares are reported. This is misleading as what they have found is that those variables are significant predictors of food addiction. If this is a misunderstanding on my part, then it would be good to clarify these section and for the authors to actually report the actual tests. Although when looking at the p values they match the ones on Table 3. 

Thank you for your comment. Table 3 shows the bivariate analysis prior to the multivariate analysis. This analysis was performed using Pearson's chi-squared test and estimating the OR (95%CI) by logistic regression. As can be seen, when the confidence interval does not contain the value 1, it coincides with the existence of a statistically significant association of the chi-squared test. On the other hand, the most relevant analysis is the multivariate analysis; for this reason we have not gone into the description of the bivariate analysis. As this is an observational study, only the multivariate analysis can control for confusion bias.

  1. Add what is meant by 'ref' in Table 3 in the note section. 

It has been added

  1. Line 180: separate section under a "discussion" heading

Thank you for your comment. It has been corrected.

  1. section 4.2: please remove the line about not being able to compare the findings of current study with other studies that have measured anxiety/depression as no other study has used that measure. If this line is kept please provide evidence that the measure that you have used for depression and anxiety is significantly operationalising those two concepts differently than other measures of depression and anxiety. Please provide a justification in introduction and method why this particular measure was used instead of others. 

Thank you for your comment. The line has been removed.

  1. Please include cronbachs alpha values where appropriate.

Unfortunately, several of the validation manuscripts of these questionnaires do not report cronbach's alpha values. Only the sleep and YFAS values are available. This is the reason why they have not been included. Following your advice, both values have been included. If you think they should not be included, please let us know.

Pittsburgh Sleep Quality Index (PSQI): 0.72

YFAS 2.0: 0.94

  1. The conclusion does not match your findings. Where in the study did you acquire evidence that food addiction is not the origin of the remaining problems but rather the consequence of the confluence of various habits? This is not what you investigated. Generally you are over estimating what you results are telling you and please take into consideration the design of the study which is one measurement in time.  

Thank you for your comment. Indeed, the design of the study does not provide evidence that food addiction is the origin or the consequence.

The conclusions have been redrafted in an attempt to clarify this issue.

  1. Overall the conclusion really needs to rewritten and reconsidered. Some of the sentences make no sense or have no actual points. 

Thank you for your comment. The conclusion has been amended. In this section the objective of the study has been connected to the results found for a better understanding.

Reviewer 4 Report

This is an interesting article examining the association between food addiction, body mass index, measures of diet, anxiety/depression, sleep, physical activity and sedentary behavior.  In a sample of nursing students the authors found significant correlations between food addiction (based on the Yale instrument) and BMI, anxiety/depression, sleep and sedentary behavior.  Further, they developed a predictive model of risk of food addiction from the data based on the correlated variables.  The authors imply (lines 68-70) that this is the first paper to examine food addiction with these variables in a non-clinical population and the first to develop a predictive algorithm for risk of food addiction (line 267).

Comments:

Line 31.  The authors cite a wide range for projected obesity prevalence by 2025 of 13%-43%.  The authors should clarify why the range is so large…e.g., it accounts for numerous countries having different prevalence projections?

Line 63.  Use people first language when referring to patients.  I.e., “patients who have overweight/obesity” instead of patients who “are” overweight/obese.

Line 71.  The authors say they assessed …”health habits, body composition and health status”.  This should be corrected to say “health habits, body mass index, and health status”, as body composition was not assessed.

Table 1.  The response rate to the alcohol question seems quite low.  Do the authors have an explanation for this low response rate compared to the other items queried?

Before Line 180.  A header for the Discussion should be inserted here.

Line 180.  The authors should say “536 nursing students” given that this is a select sample that may not represent all university students or people in general of this age category.

Lines 236-237.  The authors state that …”sleep disturbances lead to increased impulsive behaviors…”.  The authors should consider stating this as “sleep disturbances may lead to increased impulsive behaviors…” given that the preceding part of the paragraph suggests that other studies have found associations between sleep disturbances and impulsive behaviors…not providing evidence of cause and effect.

Overall purpose and application:

It is not clear what is the purpose of developing a predictive equation for food addiction from measures of sleep, BMI, depression/anxiety, and sedentary behavior?  This was not stated as a purpose in the introduction, yet it was highlighted as a unique element of this paper.  If the purpose is to diagnose food addiction from these measures, why not just use the Yale food addiction instrument?  The correlated variables also require the use of questionnaires, so there is no obvious benefit to using these assessment tools instead of the Yale instrument. 

The authors suggest that food addiction may be part of the pathogenesis of obesity (line 202), which is a major health concern.  Further, they suggest that treatment should focus on changing lifestyle behaviors, by improving sleep, quality of life and sedentary behavior (lines 214-21). If the authors main idea is that if food addiction is present there should be a different approach to treatment of obesity, then they should elaborate on that theme.  If this is the case how would treatment differ from the most effective current treatments for obesity, which include things like cognitive behavioral therapy, diet, exercise, sedentary behavior reduction and improved sleep?  In other words, what is it about food addiction that would change therapy or clinical practice…or prevention, that is different from current therapeutic approaches to overweight/obesity? 

This paper would benefit from a stronger framework tying together the purpose of the study and how the findings specifically address the purpose as well as the implications of the findings to advance the science in the field of food addiction.

Author Response

Response to reviewer #4

This is an interesting article examining the association between food addiction, body mass index, measures of diet, anxiety/depression, sleep, physical activity and sedentary behavior.  In a sample of nursing students the authors found significant correlations between food addiction (based on the Yale instrument) and BMI, anxiety/depression, sleep and sedentary behavior.  Further, they developed a predictive model of risk of food addiction from the data based on the correlated variables.  The authors imply (lines 68-70) that this is the first paper to examine food addiction with these variables in a non-clinical population and the first to develop a predictive algorithm for risk of food addiction (line 267).

Comments:

Line 31.  The authors cite a wide range for projected obesity prevalence by 2025 of 13%-43%.  The authors should clarify why the range is so large…e.g., it accounts for numerous countries having different prevalence projections?

It is drafted as follows:

In the WHO European Region, the prevalence of obesity is also increasing in most countries, with a projected prevalence of obesity of between 13% and 43% by 2025 [2]; 33 of the 53 countries are projected to have an obesity prevalence of 20% or more. Ireland was predicted to have the highest prevalence (43%) whereas Italy had the lowest projected prevalence (13%).

Line 63.  Use people first language when referring to patients.  I.e., “patients who have overweight/obesity” instead of patients who “are” overweight/obese.

Thank you for your comment. It has been corrected.

Line 71.  The authors say they assessed …”health habits, body composition and health status”.  This should be corrected to say “health habits, body mass index, and health status”, as body composition was not assessed.

Thank you for your comment. It has been corrected.

Table 1.  The response rate to the alcohol question seems quite low.  Do the authors have an explanation for this low response rate compared to the other items queried?

There was an error in writing the alcohol data. The table has been corrected.

Before Line 180.  A header for the Discussion should be inserted here.

Thank you for your comment. It has been corrected.

Line 180.  The authors should say “536 nursing students” given that this is a select sample that may not represent all university students or people in general of this age category.

Thank you for your comment. It has been corrected.

Lines 236-237.  The authors state that …”sleep disturbances lead to increased impulsive behaviors…”.  The authors should consider stating this as “sleep disturbances may lead to increased impulsive behaviors…” given that the preceding part of the paragraph suggests that other studies have found associations between sleep disturbances and impulsive behaviors…not providing evidence of cause and effect.

The sentence was incorrectly worded. It has been corrected.

Overall purpose and application:

It is not clear what is the purpose of developing a predictive equation for food addiction from measures of sleep, BMI, depression/anxiety, and sedentary behavior?  This was not stated as a purpose in the introduction, yet it was highlighted as a unique element of this paper.  If the purpose is to diagnose food addiction from these measures, why not just use the Yale food addiction instrument?  The correlated variables also require the use of questionnaires, so there is no obvious benefit to using these assessment tools instead of the Yale instrument. 

Thank you for your observation. The model developed is not intended to be a diagnostic test, it does not have that purpose. The main purpose of this predictive model is to be able to determine the risk of food addiction based on other variables. The capacity of the model is considered to be good and it can guide clinicians in the search for problems. Specifically, in the coexistence of psychological problems, sleep disturbances, sedentary lifestyle and overweight/obesity, it would be necessary to explore food addiction disorders.

The authors suggest that food addiction may be part of the pathogenesis of obesity (line 202), which is a major health concern.  Further, they suggest that treatment should focus on changing lifestyle behaviors, by improving sleep, quality of life and sedentary behavior (lines 214-21). If the authors main idea is that if food addiction is present there should be a different approach to treatment of obesity, then they should elaborate on that theme.  If this is the case how would treatment differ from the most effective current treatments for obesity, which include things like cognitive behavioral therapy, diet, exercise, sedentary behavior reduction and improved sleep?  In other words, what is it about food addiction that would change therapy or clinical practice…or prevention, that is different from current therapeutic approaches to overweight/obesity? 

This paper tries to suggest that food addiction is about behaviours. Behaviours are what make a patient obese and that obesity generates food addiction. Obesity treatment is indeed effective in managing food addiction. Our study suggests that treatments based on behavioral modifications can be effective and additional pharmacological or other treatments might not be necessary.

Conclusions section has been modified.

This paper would benefit from a stronger framework tying together the purpose of the study and how the findings specifically address the purpose as well as the implications of the findings to advance the science in the field of food addiction.

In order to try to link the objective and the findings obtained, the following text has been added to the conclusion:

“The aim of the study was to identify which lifestyle factors of nursing students were related to food addiction. Our results found no association with smoking or alcohol consumption and also no association with physical activity or Mediterranean diet. However, relationships were found with sedentary lifestyle, anxiety/depression and sleep. These lifestyle elements could be the focus of behavior modification for students.”

Thank you for all your suggestions and comments.

Round 2

Reviewer 3 Report

1.The diet assessment questionnaire citation is confusing- it takes you to a paper that looked at Food Frequency questionnaires and portions sizes but do not refer to a yes/no questionnaire. Furthermore, the PREDIMED is a trial not an actual questionnaire (all assumed from the citation the authors used). Did the authors adapt the FFQ to a yes/no scale? If yes, can you please report that. Can you also report a bit more information (including citations) when that questionnaire has been used in that way before and that it was reliable and valid.

2.In one of your conclusions you say "University nursing students, due to their training, receive knowledge regarding the effects on health or disease as a consequence of habits. This knowledge could be the reason why a lower prevalence of food addiction has been found compared to other non-clinical population groups. "  By stating this you assume that just having the knowledge can affect how people act during 'addiction' and ignore the biological basis of addiction - this in turn also makes me want to start questioning the whole term of food addiction.  You also use the word habits, when habits and addiction are two different things. I would recommend to make sure you use similar language throughout.

3. The two paragraphs from 317-323 raise the same issue so not clear why authors chose to divide into two paragraphs rather than together.

Author Response

RESPONSE TO REVIEWERS

Reviewer #3

1.The diet assessment questionnaire citation is confusing- it takes you to a paper that looked at Food Frequency questionnaires and portions sizes but do not refer to a yes/no questionnaire. Furthermore, the PREDIMED is a trial not an actual questionnaire (all assumed from the citation the authors used). Did the authors adapt the FFQ to a yes/no scale? If yes, can you please report that. Can you also report a bit more information (including citations) when that questionnaire has been used in that way before and that it was reliable and valid.

In that reference, specifically in table 1 of this article, you can find the criteria of the questionnaire for rating each question with 0 or 1 points. This was the reason why it was included in the bibliography. However, as this reference is confusing, we believe that it is better to include the following reference and delete the previous one.

Schröder, H.; Fitó, M.; Estruch, R.; Martínez‐González, M.A.; Corella, D.; Salas‐Salvadó, J.; Lamuela‐Raventós, R.; Ros, E.; Salaverría, I.; Fiol, M.; et al. A Short Screener Is Valid for Assessing Mediterranean Diet Adherence among Older Spanish Men and Women. J. Nutr. 2011, 141, 1140–1145, doi:10.3945/jn.110.135566.

Thank you very much for your comment.

 2.In one of your conclusions you say "University nursing students, due to their training, receive knowledge regarding the effects on health or disease as a consequence of habits. This knowledge could be the reason why a lower prevalence of food addiction has been found compared to other non-clinical population groups. "  By stating this you assume that just having the knowledge can affect how people act during 'addiction' and ignore the biological basis of addiction - this in turn also makes me want to start questioning the whole term of food addiction.  You also use the word habits, when habits and addiction are two different things. I would recommend to make sure you use similar language throughout.

Thank you for your comments. A better explanation has been added in the conclusions section.

  1. The two paragraphs from 317-323 raise the same issue so not clear why authors chose to divide into two paragraphs rather than together.

Thank you for your advice. Both paragraphs have been merged.

Reviewer 4 Report

Lines 12, 32, 33, 294;  use people first language...i.e., students with overweight/obesity, etc. instead of overweight/obese students or students that were overweight/obese... 

Line 68; change  have "overweight/obese" to have "overweight/obesity" 

I am still unclear about the "chicken/egg" element of their conclusion (e.g., line 306) that food addiction could be the result of obesity or poor sleep habits, etc. vs. it may be on the causal pathway (at least to obesity).  How do the authors disentangle cause and effect here?  

In the response to reviewers the authors suggest that someone presenting with overweight/obesity and perhaps also poor sleep hygiene might be at risk for food addiction.  How would this recognition change the treatment plan?  If treating obesity and poor sleep also treats food addiction, what role does suspecting food addiction play in the treatment and outcome?

To put food addiction into a more complete context, have any studies been done whereby food addiction is treated in isolation to see if obesity is reversed and healthy sleep is restored, or is food addiction always treated by treating obesity/sleep?  What is the prevalence of food addiction among non-obese, non-sleep disturbed individuals?

Author Response

RESPONSE TO REVIEWERS

Reviewer #4

Lines 12, 32, 33, 294;  use people first language...i.e., students with overweight/obesity, etc. instead of overweight/obese students or students that were overweight/obese... 

Thank you for your comment. It has been corrected.

Line 68; change  have "overweight/obese" to have "overweight/obesity" 

Thank you for your comment. It has been corrected.

I am still unclear about the "chicken/egg" element of their conclusion (e.g., line 306) that food addiction could be the result of obesity or poor sleep habits, etc. vs. it may be on the causal pathway (at least to obesity).  How do the authors disentangle cause and effect here?  

Thank you for your comments. A better explanation has been added in the conclusions section. However, the design of our study does not allow us to find the solution. With this study we cannot know for sure whether food addiction is the origin or the consequence.

In the response to reviewers the authors suggest that someone presenting with overweight/obesity and perhaps also poor sleep hygiene might be at risk for food addiction.  How would this recognition change the treatment plan?  If treating obesity and poor sleep also treats food addiction, what role does suspecting food addiction play in the treatment and outcome?

Thank you for your comments. A better explanation has been added in the conclusions section. Our proposal is to detect at an early stage those aspects of lifestyle that we have found to be related to food addiction with the predictive model. In this way, we aim to reduce the prevalence of food addiction. We did not mean that treatment would change, but that we should prevent the emergence of unhealthy lifestyles in order to avoid further complications.

To put food addiction into a more complete context, have any studies been done whereby food addiction is treated in isolation to see if obesity is reversed and healthy sleep is restored, or is food addiction always treated by treating obesity/sleep? 

Research on food addiction measured with the YFAS 2.0 instrument is very young. Therefore, there are very few studies that have conducted an intervention on food addiction and tested the results in terms of BMI and sleep. We found only one intervention study on food addiction. It was a three-month study with a control (n=25) and an intervention group (n=24). The intervention was based on motivational interviewing. There were changes over time in individual symptoms of addictive eating and dietary intakes for both groups in core and non-core foods, however, there were no significant changes in self-reported BMI.

                  Burrows, T.; Collins, R.; Rollo, M.; Leary, M.; Hides, L.; Davis, C. The feasibility of a personality targeted intervention for addictive overeating: FoodFix. Appetite 2021, 156, doi:10.1016/j.appet.2020.104974.

There are very few studies (and no intervention studies) dealing with food addiction and sleep.

What is the prevalence of food addiction among non-obese, non-sleep disturbed individuals?

Here are the results for food addiction in the sample of students we have analysed:

Students with good sleep and no overweight/obesity: 1.1%

Students with good sleep and overweight/obesity: 11.1%

Students with bad sleep and no overweight/obesity: 6%

Students with bad sleep and overweight/obesity: 34.3%

Thank you very much for all your comments and advice.